# The Impact of Prescription Time Limits on Phosphate Administration in the Intensive Care Unit: A Before–After Quality Improvement Study

**DOI:** 10.3390/healthcare12151549

**Published:** 2024-08-05

**Authors:** Rajiv Rooplalsingh, Felicity Edwards, Julia Affleck, Patrick Young, Alexis Tabah, Sinead Carmichael, Belinda Chappell, Andrea Fung, Kylie Jacobs, Kevin Laupland, Mahesh Ramanan

**Affiliations:** 1Department of Anaesthesia, Royal Brisbane and Women’s Hospital, Brisbane, QLD 4029, Australia; rajivricu@gmail.com; 2Faculty of Health, Queensland University of Technology (QUT), Brisbane, QLD 4000, Australia; felicity.edwards@qut.edu.au (F.E.); alexis@tabah.org (A.T.); kevin.laupland@qut.edu.au (K.L.); 3Department of Intensive Care Services, Royal Brisbane and Women’s Hospital, Brisbane, QLD 4029, Australia; 4Research Development Unit, Caboolture Hospital, Caboolture, QLD 4510, Australia; julia.affleck@health.qld.gov.au; 5Intensive Care Unit, Redcliffe Hospital, Redcliffe, QLD 4020, Australia; patrick.young@health.qld.gov.au (P.Y.); kylie.jacobs@health.qld.gov.au (K.J.); 6Intensive Care Unit, Caboolture Hospital, Caboolture, QLD 4510, Australia; 7School of Medicine, The University of Queensland, Brisbane, QLD 4000, Australia; 8Pharmacy Department, Royal Brisbane and Women’s Hospital, Brisbane, QLD 4029, Australia; sinead.carmichael@health.qld.gov.au; 9Pharmacy Department, Caboolture Hospital, Caboolture, QLD 4510, Australia; belinda.chappell@health.qld.gov.au; 10Pharmacy Department, Redcliffe Hospital, Redcliffe, QLD 4020, Australia; andrea.fung@health.qld.gov.au; 11Intensive Care Unit, The Prince Charles Hospital, Brisbane, QLD 4032, Australia

**Keywords:** phosphate, electrolytes, critical care, enteral, intravenous, quality improvement

## Abstract

(1) Background: We aim to examine and improve phosphate prescribing as part of a quality assurance program by examining the change in the proportion of patients receiving phosphate with normal or high preceding serum phosphate concentrations before and after the introduction of the 24 h time limit to default phosphate prescription. (2) Methods: This was a quality assurance study conducted across three Australian adult intensive care units (ICUs). All adult patients with ICU lengths of stay greater than or equal to 48 h who had their serum phosphate concentrations measured were included. A 24 h time limit was introduced to the protocolised prescription in the electronic clinical information system for enteral and intravenous phosphate at participating ICUs. Patient characteristics, phosphate administration, and outcomes were compared before and after this time limit was introduced. The primary outcome was the proportion of patients to whom phosphate was prescribed after measurement of a normal or high serum phosphate level. Secondary outcomes were ICU length of stay, mortality, and discharge destination. (3) Results: A total of 1192 patients were included from three ICUs over the two periods. The proportion of patients with a normal or high measured phosphate level who then received phosphate supplementation was significantly lower in the second study period (30.3% vs. 9.9%; *p* < 0.001). This difference persisted when adjusted for potential confounders in a mixed-effects logistic regression model (an adjusted odds ratio for receiving phosphate with normal or high serum concentration 0.214, 95% confidence interval of 0.132–0.347; *p* < 0.001). No significant difference was seen in the typical ICU length of stay, in-hospital case–fatality rate, and hospital discharge destination between these groups. (4) Conclusions: This multicentre before–after study has demonstrated that the introduction of a 24 h limit on electronic phosphate prescriptions resulted in significantly fewer patients receiving phosphate when their serum phosphate concentration was normal or high, without any adverse impact on patient outcomes.

## 1. Introduction

Phosphate is a predominately intracellular anion essential for life, the majority of which exists in bone. A minority, 15%, is distributed as ATP (adenosine triphosphate) and cellular structural proteins (phospholipids, nucleic acids and phosphoproteins). Its presence as an anion functions as a buffer both intracellularly and as an extracellular urinary buffer. Potential deleterious effects of hypophosphataemia include paresthesia, muscle weakness, rhabdomyolysis, cardiac failure, seizures, and coma. Serum phosphate concentrations typically range between 0.80 and 1.50 mmol/L, and endocrine control of phosphate concentration is closely linked to calcium homeostasis through the regulatory substances parathyroid hormone (PTH), vitamin D, fibroblast growth factor 23 (FGF23), and co-factor glucuronidase klotho [1,2,3].

Critical illness requiring admission to the intensive care unit (ICU) frequently results in unstable phosphate levels. Prior studies have demonstrated incidences of hyperphosphatemia and hypophosphataemia of 19.3% and 10.8–15.4%, respectively [4,5]. The literature is conflicting with regards to the effect of hypophosphataemia on patient outcomes, with one retrospective study demonstrating an association with an increased risk of death (OR 1.29; 95% CI, 1.02–1.64; *p* = 0.034) and a recent systemic review demonstrating and an increased average ICU length of stay (2.22 days [95% CI, 1.00–3.44]) but no association with increased mortality (risk ratio: 1.13 [95% CI, 0.98–1.31]; *p* = 0.09) [5,6]. Conversely there is a clear association between hyperphosphatemia and increased risk of death. Two recent retrospective studies, one including ICU patients demonstrating an increased risk of death (OR 1.39; 95% CI, 1.15–1.68; *p* = 0.001) and a second study of hospital patients (excluding those in ICU) also demonstrating an increased risk of death (16.4 vs. 6.6%, *p* < 0.001) [7,8], have proven this link. Despite this, phosphate in either intravenous or enteral preparations is often administered to patients in ICU at a rate of 41.8–57.1% [4,5,6,7,8,9]. In addition to this, a treatment protocol existed in only 41.1% of adult ICUs [4].

An audit of phosphate prescriptions was conducted as part of routine quality assurance activities at three ICUs (at which the authors are employed). This occurred within a health service provider in the state of Queensland, Australia, and revealed that up to a third of patients with normal or high serum phosphate concentrations (i.e., patients who should generally not be administered phosphate) were receiving phosphate. A simple change, which entailed the introduction of a 24 h limit on the default phosphate prescriptions entered into the electronic clinical information system in use at all three ICUs was introduced in response.

In this study, we describe the effect of this simple change on phosphate administration to patients with normal or high preceding serum phosphate concentrations. We hypothesised that introducing a 24 h limit on default phosphate prescriptions in the electronic clinical information system would result in the reduced administration of phosphate to patients with normal or high phosphate concentrations.

## 2. Materials and Methods

### 2.1. Study Design and Patients

This is a cohort study; a before–after quality improvement study was conducted at three adult ICUs, located at Royal Brisbane and Women’s Hospital, Redcliffe Hospital, and Caboolture Hospital, all part of the Metro North Hospital and Health Services provider, Queensland, Australia. Patients admitted to participating ICUs with lengths of stay greater than 48 h who had at least one serum phosphate measurement were included over two distinct time periods: January to August 2023 (Period one) and October to November 2023 (Period two). The initial phosphate level could be measured at any time during the patient’s intensive care unit stay. Subsequent measurements were at the discretion of the treating clinicians. Once recruited, the patients participated in this study for the duration of their intensive care unit stay. However, for the purposes of statistical analysis, each patient was included once only, regardless of the number of phosphate measurements or doses that they received for that intensive care admission.

### 2.2. Data Sources and Variables

Information regarding study participants and their characteristics was identified using the eCritical MetaVision^TM^ 5.4 (iMDsoft, Boston, MA, USA) clinical information system. This included demographic data (age, gender), co-morbidities (chronic cardiovascular/respiratory/liver/renal/immunosuppressive disease, metastatic cancer and diabetes), admission diagnosis (elective surgery, sepsis, trauma), Acute Physiology and Chronic Health Evaluation—Two Score (APACHE-II) [10], frailty score using the Clinical Frailty Score (CFS) [11,12], Australian and New Zealand Risk of Death score (ANZROD) [13], ICU organ supports (invasive ventilation and renal replacement therapy), serum phosphate levels, and the number of phosphate supplement doses received.

The APACHE-II score uses point values allocated to 12 physiological measurements, age, and chronic health state to provide a general measure of disease severity. The physiological variables included are the most unstable values recorded over the first 24 h of the patient’s ICU stay. They include temperature, mean arterial pressure, heart rate, respiratory rate, oxygenation, arterial pH, serum sodium, serum potassium, serum creatinine, hematocrit, and white blood count. Chronic health state includes chronic cardiovascular/respiratory/liver/renal disease and immunosuppressive state. Definitions for all data points can be found in the Australia and New Zealand Intensive Care Society Adult Patient Database Data Dictionary [14].

### 2.3. Intervention

All three ICUs used the same phosphate replacement protocol administered through a clinical information system, MetaVision (See Appendix A for the specifics of the protocol). In brief, phosphate replacement commenced once serum phosphate concentration fell below 0.80 mmol/L, with higher doses being administered for lower concentrations. The enteral versus intravenous route was decided based on whether the patient had a functioning enteral tract or not (determined by treating clinician). The use of the phosphate replacement protocol was highly encouraged but not mandated.

For study period one, phosphate prescriptions by default were not time limited. It was at the discretion of the treating clinician to initiate the standardised phosphate prescription, which did not include a duration of validity. Thus, the prescription lasted for the duration of the patient’s admission to the ICU. At the completion of study period one, the intervention was introduced. 

For study period two, a 24 h time limit was applied to the default phosphate prescription. Upon the admission of a patient to the intensive care unit, it was again at the discretion of the treating clinician to initiate the standardised phosphate prescription, which for study period two was valid for 24 h. This meant that the prescription was automatically terminated 24 h after its initiation. Clinicians then had to re-prescribe phosphate if required. 

For both the study periods mentioned above, clinicians could also choose to alter the default prescription and increase or remove the time limit. Thus, the process of care was changed, but the clinician’s ability to make clinical decisions based on their judgement was not changed.

All ICUs used standard phosphate preparations included on the List of Approved Medications used by all Queensland Health facilities. These were Phosphate Sandoz (Novartis Pharmaceuticals Australia Pty Limited, North Ryde, New South Wales, Australia), containing 500 mg (16.1 mmol) of elemental phosphate for enteral use, and sodium dihydrogen phosphate (10 mmol elemental phosphorus; Phebra Pty Ltd., Lane Cove West, NSW, Australia) for intravenous use.

Patients to whom supplemental phosphate was administered were stratified on the basis of their pre-dose serum concentration. Strata were as follows; ≤0.79, 0.8–1.5, and >1.5 mmol/L, representing hypophosphatemia, normal, and hyperphosphatemia, respectively [6].

The primary outcome was the proportion of patients with normal or high preceding phosphate levels to whom supplemental phosphate was administered. The exposure variable was the time period either before (study period one) or after (study period two) the introduction of the 24 h time limit on phosphate prescriptions.

Secondary outcomes were the ICU length of stay (days) and hospital outcome (died in hospital or discharged to one of the following destinations: home, nursing home, other ICU, other hospital, rehabilitation unit, mental health unit, or hospital in the home).

### 2.4. Statistical Analysis

Continuous and categorical data were presented as the median and interquartile range (IQR) or frequencies and proportions. The Mann–Whitney U test and the Pearson chi-squared test were used for univariate comparisons. Mixed-effects logistic regression modelling was used to evaluate the effect of the intervention on the prescription of phosphate to patients with normal or high serum phosphate concentrations. Age, Australia and New Zealand Risk of Death, elective surgery, invasive ventilation on day 1 of ICU admission, and receipt of renal replacement therapy during ICU admission were entered as fixed effects, with the site denoted as a random effect, and patients nested within sites. Our results were presented as odds ratio and 95% confidence intervals, with a *p*-value < 0.05 being statistically significant. All analyses were performed in Stata 17.0 (StataCorp, College Station, TX, USA).

### 2.5. Research Ethics

This study was conducted in accordance with the Declaration of Helsinki, and exemption from ethics approval under quality assurance provisions, in compliance with Australian Federal and State laws, was obtained from the Metro North Human Research Ethics Committee “A” [EX/2024/MNHA/105985].

## 3. Results

Over the two study periods, 3204 patients were admitted to the three ICUs, of whom 1245 met the inclusion criteria of an ICU length of stay >48 h. Of these 1245 patients, 53 were excluded as serum phosphate was not measured. A total of 1192 patients were included in the analysis, with 989 and 203 in periods one and two, respectively (Figure 1 and Table 1 and Table 2). In period one, 300 (30.3%) patients received phosphate with a normal or high preceding serum phosphate concentration, compared to 20 (9.9%) patients in period two (Table 2). 

Patient characteristics were similar overall between the study periods with regard to age; sex; chronic cardiovascular, respiratory, liver, renal, and immunosuppressive disease; diabetes, frailty (clinical frailty scale), APACHE-II score, and ANZROD (Australia New Zealand Risk of Death), as well as regarding the proportion of patients admitted for sepsis and trauma. There were more patients in period two with metastatic cancer (10 (4%) vs. 18 (1.8%); *p* = 0.041) and elective surgical admission (28 (11.1%) vs. 64 (6.5%); *p* = 0.012) and fewer with invasive ventilation on day 1 of ICU admission (114 (45.1%) vs. 568 (57.3%); *p* < 0.001) (Table 1).

Across all hospitals, a lower proportion of patients with a preceding normal or high phosphate level received phosphate in period two compared to period one, which reached statistical significance in Hospital A and the overall analysis (Table 2). The adjusted analysis confirmed that patients in period two were significantly less likely to be administered phosphate with a normal or high serum phosphate concentration (adjusted odds ratio 0.214; 95% confidence interval 0.132–0.347; *p* < 0.001) (Table 3). There were no differences in patient outcomes for measures including died in hospital, ICU length of stay, and hospital discharge destination (home, nursing home, other ICU, other hospital, rehabilitation unit, mental health unit, or hospital in the home) between the two study periods (Table 4).

## 4. Discussion

This multicentre before–after quality improvement study demonstrates that the simple addition of a 24 h time limit to the phosphate prescription in a clinical information system resulted in a significant decrease in the proportion of patients receiving phosphate supplementation with normal and high preceding phosphate levels. As shown in Table 2, this difference was 30.3% vs. 9.9%, with *p* < 0.001, for periods one and two, respectively. The difference persisted after adjusting for covariables using logistic regression analysis. These results suggest that a simple change to the default prescription is both feasible and effective at significantly altering unnecessary phosphate administration.

As stated previously, admission to ICU is frequently associated with abnormal phosphate levels, itself resulting in attributable mortality, with the conditions with which it is associated, thus, serving as a marker of severity of illness. These results are important in the context of a prior study demonstrating an association between hyperphosphatemia and an increased risk of death [7]. While the association between phosphate levels and risk of death remains controversial and unresolved, phosphate administration has not been demonstrated to unequivocally improve patient outcomes either [5,7]. Thus, on both counts, reducing unnecessary phosphate administration to patients with normal serum concentrations is appropriate. Further, while the reduction in phosphate administration to patients with hyperphosphatemia was not statistically significant, from a patient safety perspective, it is notable that there were no patients at all in period two who received phosphate while being hyperphosphataemic.

In addition to this are the variable results of trials investigating the utility of computerised decision support systems in reducing drug errors. Numerous barriers to their implementation exist, with some studies demonstrating that over half of alerts were ignored by prescribers [15,16,17]. The 24 h time limit bypasses this problem as clinicians have to either modify the default prescription or re-prescribe to intentionally deliver longer courses of phosphate. 

The intervention was free of cost, extremely simple to introduce, and applicable to all patients admitted to the ICU. 

There were some limitations to this study. As it was a before–after quality improvement study, caution must be taken in making claims of causal inference. While appropriate techniques were used to address some confounders, the possibility of residual confounding remains. This study was performed using routinely collected electronic health record data, and as such, clinical reasoning for decisions made by clinicians with regard to phosphate prescriptions was not outlined. Two distinct time periods were used, period one and period two, lasting eight and two months, respectively. There was a small amount of missing data. While this study did not demonstrate any significant differences in patient outcomes, it was not designed to detect these differences, nor is there biological rationale to suspect that outcomes would be different according to phosphate replacement. Further, large-scale implementation studies designed to detect small changes in patient outcomes may be useful as part of ongoing quality assurance processes. 

## 5. Conclusions

This multicentre before–after quality improvement study has demonstrated that the prescription of phosphate to ICU patients with normal or high preceding serum phosphate concentrations occurs at an alarming rate. The number of prescriptions can be effectively reduced with the introduction of a 24 h time limit on phosphate prescriptions within an electronic clinical information system. As this was not a randomised control study, association but not causation could be inferred. A potential limitation of this catch-all approach may be that the necessity of re-prescribing subsequent doses may increase workloads for treating clinicians. Institutions without a clinical information system may find this approach difficult to implement.

## Figures and Tables

**Figure 1 healthcare-12-01549-f001:**
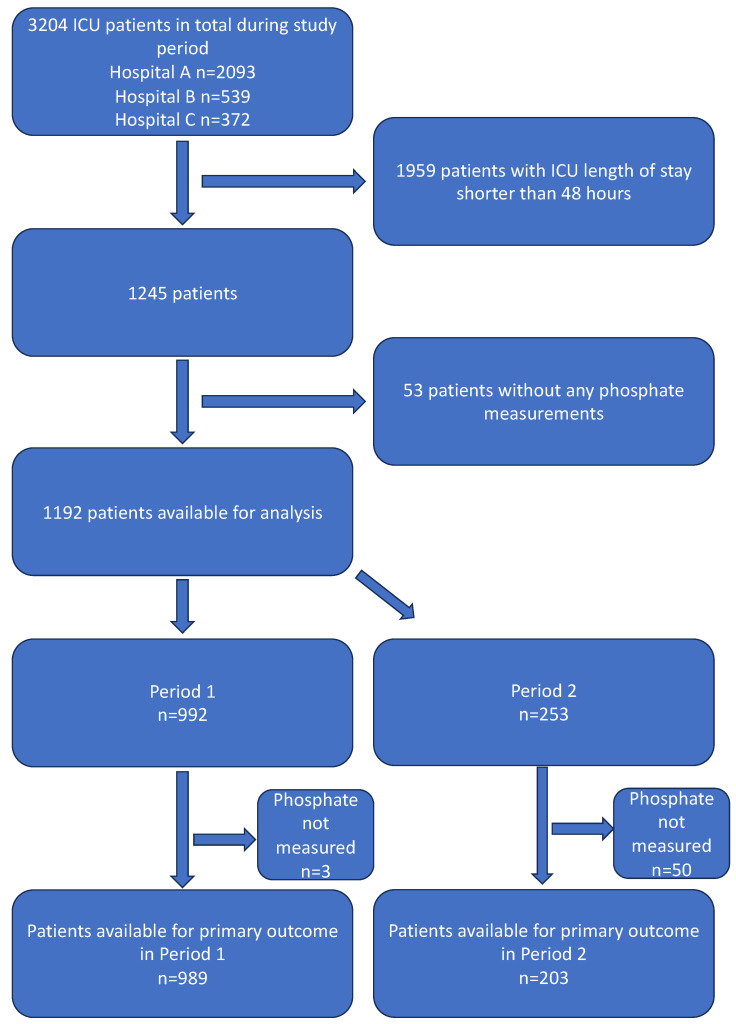
Patient recruitment diagram (CONSORT diagram).

**Table 1 healthcare-12-01549-t001:** Patient baseline characteristics.

	Period 1	Period 2	Total	*p*-Value
	N = 992	N = 253	N = 1245	
Age	60.4 (44.95–71.5)	59.8 (47.2–71)	60.3 (45.1–71.4)	0.6
Female sex	436 (44.0%)	107 (42.3%)	543 (43.6%)	0.63
Chronic cardiovascular disease	41 (4.1%)	8 (3.2%)	49 (3.9%)	0.48
Chronic respiratory disease	71 (7.2%)	13 (5.1%)	84 (6.7%)	0.25
Chronic liver disease	36 (3.6%)	5 (2.0%)	41 (3.3%)	0.19
Chronic renal disease	21 (2.1%)	7 (2.8%)	28 (2.2%)	0.53
Chronic immunosuppressive disease	65 (6.6%)	22 (8.7%)	87 (7.0%)	0.23
Metastatic cancer	18 (1.8%)	10 (4.0%)	28 (2.2%)	0.041
Diabetes				0.26
Other ^f^	7 (0.7%)	1 (0.4%)	8 (0.6%)	
Type 1	13 (1.3%)	4 (1.6%)	17 (1.4%)	
Type 2	214 (21.6%)	41 (16.2%)	255 (20.5%)	
Clinical FRAILTY Scale				0.52
1	62 (6.3%)	11 (4.3%)	73 (5.9%)	
2	224 (22.6%)	55 (21.7%)	279 (22.4%)	
3	201 (20.3%)	62 (24.5%)	263 (21.2%)	
4	214 (21.6%)	51 (20.2%)	265 (21.3%)	
5	103 (10.4%)	23 (9.1%)	126 (10.1%)	
6	132 (13.3%)	32 (12.6%)	164 (13.2%)	
7	43 (4.3%)	17 (6.7%)	60 (4.8%)	
8	11 (1.1%)	2 (0.8%)	13 (1.0%)	
Invasively ventilated on day 1	568 (57.3%)	114 (45.1%)	682 (54.8%)	<0.001
APACHE-2 score	18 (14–24)	18 (14–23)	18 (14–23)	0.33
ANZROD	0.067 (0.023–0.217)	0.076 (0.02–0.175)	0.068 (0.022–0.201)	0.43
Elective surgery	64 (6.5%)	28 (11.1%)	92 (7.4%)	0.012
Sepsis admission	117 (11.8%)	30 (11.9%)	147 (11.8%)	0.98
Trauma admission	135 (13.6%)	29 (11.5%)	164 (13.2%)	0.37

Data are presented as median (IQR) for continuous measures and n (%) for categorical measures. The following chronic disease states form the basis of the APACHE-II score: cardiovascular disease, respiratory disease, liver disease, renal disease, immunosuppressive disease, and metastatic cancer [13]. Clinical frailty scale based on the 9-point scale (1: Very fit; 2: Fit; 3: Managing well; 4: Living with very mild frailty; 5: Living with mild frailty; 6: Living with moderate frailty; 7: Living with severe frailty; 8: Living with very severe frailty; 9: Terminally ill), updated version 2.0 [14,15]. ^f^ Secondary diabetes, previous gestational diabetes, and impaired fasting glucose/glucose intolerance. Admission diagnoses included elective surgery, sepsis admission, and trauma admission.

**Table 2 healthcare-12-01549-t002:** Phosphate administration to patients with normal or high serum phosphate levels.

Serum Phosphate Level at Which Supplemental Phosphate Was Administered	Period 1	Period 2	*p*-Value
Hospital A	n = 616	n = 136	
Normal phosphate level (0.80–1.49 mmol/L)	210 (34.1%)	11 (8.1%)	<0.001
High phosphate level (≥1.5 mmol/L)	13 (2.1%)	0 (0%)	0.087
Normal OR high phosphate level (≥0.80 mmol/L)	210 (34.1%)	11 (8.1%)	<0.001
Hospital B	n = 228	n = 36	
Normal phosphate level (0.80–1.49 mmol/L)	57 (25%)	5 (13.9%)	0.144
High phosphate level (≥1.5 mmol/L)	2 (0.9%)	0 (0%)	0.573
Normal OR high phosphate level (≥0.80 mmol/L)	59 (25.9%)	5 (13.9%)	0.119
Hospital C	n = 145	n = 31	
Normal phosphate level (0.80–1.49 mmol/L)	29 (20%)	4 (12.9%)	0.358
High phosphate level (≥1.5 mmol/L)	4 (2.8%)	0 (0%)	0.35
Normal OR high phosphate level (≥0.80 mmol/L)	31 (21.4%)	4 (12.9%)	0.283
Total—All patients with a measured phosphate level	n = 989	n = 203	
Normal phosphate level (0.80–1.49 mmol/L)	296 (29.9%)	20 (9.9%)	<0.001
High phosphate level (≥1.5 mmol/L)	19 (1.9%)	0 (0%)	0.047
Normal OR high phosphate level (≥0.80 mmol/L)	300 (30.3%)	20 (9.9%)	<0.001

This table shows the number and percentage of patients who received phosphate with their preceding phosphate level falling into the listed ranges, e.g., there were 145 patients in total at Hospital C during Period 1, and 29 (20%) patients received phosphate replacement when their most recent serum phosphate level was 0.8–1.49 mmol/L.

**Table 3 healthcare-12-01549-t003:** Primary outcome: the administration of phosphate when the serum phosphate level was greater than or equal to 0.80 mmol/L.

	Odds Ratio (95% Confidence Interval)	*p*-Value
Univariable analysis ^1^		
Period 2 (compared to Period 1)	0.195 (0.121–0.314)	<0.001
Multivariable analysis ^2^		
Period 2 (compared to Period 1)	0.214 (0.132–0.347)	<0.001
Age (per 1 year)	1 (0.992–1.008)	0.947
ANZROD ^3^ (per 1 percentage)	1.151 (0.536–2.469)	0.719
Elective surgery (compared to non-elective admission)	0.604 (0.323–1.129)	0.114
Invasively ventilated on day 1 of ICU ^4^ admission	1.626 (1.2–2.204)	0.002
Renal replacement therapy	0.818 (0.52–1.287)	0.385

^1^ This was a mixed-effects logistic regression analysis, with the period (2 compared to 1) being the fixed effect and the site being the random effect, with patients nested within sites. ^2^ This was a mixed-effects multivariable logistic regression analysis, with period (2 compared to 1) being the primary exposure variable, while age, ANZROD, elective surgery, invasive ventilation on day 1, and renal replacement therapy were the other fixed effects, and site was the random effect, with patients nested within sites. ^3^ Australia and New Zealand Risk of Death. ^4^ Intensive care unit.

**Table 4 healthcare-12-01549-t004:** Patient outcomes by period.

	Period 1	Period 2	Total	*p*-Value
	N = 992	N = 253	N = 1245	
ICU Length of stay (days)	4.15 (2.88–7.52)	4.01 (2.8–7.3)	4.11 (2.86–7.48)	0.31
Died in hospital	172 (17.3%)	45 (17.9%)	217 (17.5%)	0.76 *
Discharged home	498 (50.2%)	129 (51.4%)	627 (50.4%)	
Discharged to nursing home	9 (0.9%)	2 (0.8%)	11 (0.9%)	
Discharged to other ICU	10 (1.0%)	0 (0.0%)	10 (0.8%)	
Discharged to other hospital	175 (17.6%)	38 (15.1%)	213 (17.1%)	
Discharged to rehabilitation unit	111 (11.2%)	32 (12.7%)	143 (11.5%)	
Discharged to mental health unit	8 (0.8%)	3 (1.2%)	11 (0.9%)	
Discharged to “hospital in the home”	9 (0.9%)	2 (0.8%)	11 (0.9%)	

Data are presented as median (IQR) for continuous measures and n (%) for categorical measures. * Category *p*-value from the Pearson chi-squared test for hospital outcomes by period. The *p*-value for hospital outcomes is a group *p*-value.

## Data Availability

Data cannot be shared publicly due to institutional ethics, privacy concerns, and confidentiality regulations. Data being released for the purposes of research under Section 280 of the Public Health Act 2005 requires an application to the Director-General of Queensland Health (PHA@health.qld.gov.au).

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
