# Peer review of "The Impact of Prescription Time Limits on Phosphate Administration in the Intensive Care Unit: A Before–After Quality Improvement Study"

_healthcare, 2024, doi:10.3390/healthcare12151549_

Round 1
Reviewer 1 Report
Comments and Suggestions for Authors
Thank you for the submission, this article is interesting, but there are some comments. please see in the attachment.

Reviewer 2 Report
Comments and Suggestions for Authors
Dear Authors,
After thoroughly examining your manuscript, I would like to express my highly positive feedback regarding your submission. Your study holds significance for the readership of the “Healthcare” journal. I believe there is room for some technical improvements, and I would like to request revisions based on the following remarks:
· You indicated that your study is of a “quality assurance study” type. However, in the guide for authors such type of publication within a “Healthcare” journal cannot be found. Therefore, please consult the official guide and, if necessary, correct the type of your manuscript.
· Please indicate which software package was used for statistical analysis.
· In Table 4, please write “p-value” so that it fits in a single row.
· The conclussion should be written better. I agree with your conclusions, but this part of your manuscript should present the importance of your results in a better way. Please “strengthen” this part of your manuscript, and indicate what are the challenges and limits of your research.
· Schematics presented in Appendix A1 should be placed in a separate Supplementary file, with necessary explanations.
· It would be highly beneficial for this manuscript if a richer literature survey had been performed. The bibliography contains only 15 references.
Upon addressing all of the aforementioned comments, I would be pleased to review your manuscript again.
Best regards
Reviewer 3 Report
Comments and Suggestions for Authors
In this study, the authors describe the effect of a simple change on phosphate administration to patients with normal or high preceding serum phosphate concentrations. The study have been carried out methodically and give insightful results.
There are still some issues to be addressed:
1. The time period of Period 1 and Period 2 are widely varying. Authors may have to mention it as a limitation.
2. From, materials and methods, it is not clear how and when the 24-hour limit is applied in the patient treatment protocol. Authors may have to clearly mention how the 24-hour limit affect the decision making in the context of protocol followed. Flowchart of diagrams would be more apt in this context. What is the frequency and time duration of seum phosphate estimation and how does 24-hour limit applies to that?
3. Do the study period for each patient applies only 48 hour post admission ? Does a Normal or high phosphate level after 48 hour is also take into consideration in Table 2 ?
4. In view of the varying group sizes , authors are advised to calculate the power of the study for confirmation.
5. The Table2 and table 3 and its significance have to be discussed in detail in discussion for better understanding.
Round 2
Reviewer 3 Report
Comments and Suggestions for Authors
All comments have been adequately addressed by the authors. Additional revisions are not necessary and the manuscript can be accepted in the present form.
Author Response
Thank you for your comments.